

# Prognostic analysis of histopathological images using pre-trained convolutional neural networks: application to hepatocellular carcinoma

Liangqun Lu and Bernie J. Daigle, Jr

Departments of Biological Sciences and Computer Science, The University of Memphis, Memphis, TN, USA

## ABSTRACT

Histopathological images contain rich phenotypic descriptions of the molecular processes underlying disease progression. Convolutional neural networks, state-of-the-art image analysis techniques in computer vision, automatically learn representative features from such images which can be useful for disease diagnosis, prognosis, and subtyping. Hepatocellular carcinoma (HCC) is the sixth most common type of primary liver malignancy. Despite the high mortality rate of HCC, little previous work has made use of CNN models to explore the use of histopathological images for prognosis and clinical survival prediction of HCC. We applied three pre-trained CNN models—VGG 16, Inception V3 and ResNet 50—to extract features from HCC histopathological images. Sample visualization and classification analyses based on these features showed a very clear separation between cancer and normal samples. In a univariate Cox regression analysis, 21.4% and 16% of image features on average were significantly associated with overall survival (OS) and disease-free survival (DFS), respectively. We also observed significant correlations between these features and integrated biological pathways derived from gene expression and copy number variation. Using an elastic net regularized Cox Proportional Hazards model of OS constructed from Inception image features, we obtained a concordance index (C-index) of 0.789 and a significant log-rank test ($p = 7.6E-18$). We also performed unsupervised classification to identify HCC subgroups from image features. The optimal two subgroups discovered using Inception model image features showed significant differences in both overall ($C$-index $= 0.628$ and $p = 7.39E-07$) and DFS ($C$-index $= 0.558$ and $p = 0.012$). Our work demonstrates the utility of extracting image features using pre-trained models by using them to build accurate prognostic models of HCC as well as highlight significant correlations between these features, clinical survival, and relevant biological pathways. Image features extracted from HCC histopathological images using the pre-trained CNN models VGG 16, Inception V3 and ResNet 50 can accurately distinguish normal and cancer samples. Furthermore, these image features are significantly correlated with survival and relevant biological pathways.

Corresponding author
Bernie J. Daigle, Jr,
bjdaigle@memphis.edu

# INTRODUCTION

Histopathological images contain rich phenotypic descriptions of the molecular processes underlying disease progression and have been used for diagnosis, prognosis, and subtype discovery (*Gurcan et al., 2009*). These images contain visual features such as nuclear atypia, mitotic activity, cellular density, tissue architecture and higher-order patterns, which are typically examined by pathologists to diagnose and grade lesions. The recent accumulation of scanned and digitized whole slide images (WSIs) has enabled wide application of machine learning algorithms to extract useful information and assist in lesion detection, classification, segmentation, and image reconstruction (*Komura & Ishikawa, 2018*).

Deep learning is a machine learning method based on deep neural networks that has been widely applied in recent computer vision and natural language processing tasks (*LeCun, Bengio & Hinton, 2015*). A convolutional neural network (CNN), a class of deep learning architecture commonly used in computer vision, automatically learns representative features from images. CNNs have been dominant since their astonishing results at the ImageNet Large Scale Visual Recognition Competition (*Krizhevsky, Sutskever & Hinton, 2012*). In various studies, CNNs have shown good performance when applied to medical images, including those from radiology (*Yamashita et al., 2018*; *Shen, Wu & Suk, 2017*; *Xu et al., 2017*). Additional applications of CNNs in the areas of diabetic retinopathy screening (*Gulshan et al., 2016*), skin lesion classification (*Esteva et al., 2017*), age-related macular degeneration diagnosis *Yoo et al. (2019)* and lymph node metastasis detection (*Bejnordi et al., 2017*) have demonstrated expert-level performance in these tasks. In addition, a recent study applied CNN models to develop a content-based histopathological image retrieval tool for improving search efficiency of large histopathological image has archived (*Hegde et al., 2019*). Compared with traditional machine learning techniques, CNNs have achieved significantly improved performance in the areas of image registration for localization, detection of anatomical and cellular structures, tissue segmentation, and computer-aided disease prognosis and diagnosis (*Litjens et al., 2017*).

One disadvantage of CNNs is their need for massive amounts of data, which can be a challenge for biomedical image analysis studies. Furthermore, deep feature learning depends on the size and degree of annotation of images, which are often not standardized across different datasets. One possible solution for analyzing image datasets with a small sample size is transfer learning, in which pre-trained CNN models from large-scale natural image datasets are applied to solve biomedical image tasks. In a previous study of CNN models applied to both thoraco-abdominal lymph node detection and interstitial lung disease classification, transfer-learning from large scale annotated image datasets (ImageNet) was consistently beneficial in both tasks (*Shin et al., 2016*). Furthermore, in a

breast cancer study (*Dabeer, Khan & Islam, 2019*), CNNs used for feature extraction followed by supervised classification achieved 99.86% accuracy for the positive class.

The overarching goal of this work is to evaluate the potential of transfer learning for histopathological image analysis of hepatocellular carcinoma (HCC). Primary liver cancer is the sixth most common liver malignancy, with a high mortality and morbidity rate. HCC is the representative type, resulting from the malignant transformation of hepatocytes in a cirrhotic, non-fibrotic, or minimally fibrotic liver (*Llovet et al., 2016*). With the development of high-throughput technologies, a number of "omics" research studies have helped elucidate the mechanisms of HCC molecular pathogenesis, which in turn have significantly contributed to our understanding of cancer genomics, diagnostics, prognostics, and therapeutics (*Guichard et al., 2012*; *Totoki et al., 2011*; *Schulze et al., 2015*; *Ally et al., 2017*). In particular, the most frequent mutations and chromosome alterations leading to HCC were identified in the TERT promoter as well as the CTNNB1, TP53, AXIN1, ARID1A, NFE2L2, ARID2 and RPS6KA3 genes (*Ally et al., 2017*). The biological pathways Wnt/β-catenin signaling, oxidative stress metabolism, and Ras/mitogen-activated protein kinase were reported to be involved in liver carcinogenesis (*Guichard et al., 2012*). Frequent TP53-inactivating mutations, higher expression of stemness markers (KRT19 and EPCAM) and the tumor marker BIRC5, and activated Wnt and Akt signaling pathways were also reported to associate with stratification of HCC samples (*Ally et al., 2017*). The histological subtypes of HCC have been shown relate to particular gene mutations and molecular tumor classification (*Calderaro et al., 2017*). Two recent studies have demonstrated strong connections between molecular changes and disease phenotypes. In a meta-analysis of 1,494 HCC samples, consensus driver genes were identified that showed strong impacts on cancer phenotypes (*Chaudhary et al., 2018*). In addition, a deep learning-based multi-omics data integration study produced a model capable of robust survival prediction (*Chaudhary et al., 2017*). These and other recent findings may help to translate our knowledge of HCC biology into clinical practice (*Calderaro et al., 2017*).

At the pathological level, HCC exhibits as a morphologically heterogeneous tumor. Although HCC neoplastic cells often grow in cords of variable thickness lined by endothelial cells mimicking the trabeculae and sinusoids of normal liver, other architectural patterns are frequently observed and numerous cytological variants recognized. Though histopathologic criteria for diagnosing classical, progressed HCC are well established and known, it is challenging to detect increasingly small lesions in core needle biopsies during routine screenings. Such lesions can be far more difficult to distinguish from one another than progressed HCC, which is usually diagnosed in a straightforward manner using hematoxylin and eosin staining (*Kojiro, 2005*; *Schlageter et al., 2014*). Although prognostication increasingly relies on genomic biomarkers that measure genetic alterations, gene expression changes, and epigenetic modifications, histology remains an important tool in predicting the future course of a patient's disease. Previous studies (*Cheng et al., 2017*; *Mobadersany et al., 2018*) indicated the complementary nature of information provided by histopathological and genomic data.

Quantitative analysis of histopathological images and their integration with genomics data require innovations in integrative genomics and bioimage informatics.

In this study, we applied pre-trained CNN models on HCC histopathological images to extract image features and characterize the relationships between images, clinical survival and biological pathways. We first downloaded Hematoxylin and Eosin (H&E) stained WSIs from HCC subjects (421 tumor samples and 105 normal tissue adjacent to tumor samples) from the National Cancer Institute Genomic Data Commons Data Portal. After image normalization, we applied three pre-trained CNN models–VGG 16, Inception V3 and ResNet 50—to extract representative image features. Using these features, we (1) performed classification between cancer and normal samples, (2) constructed models associating image features with clinical survival, (3) discovered potential HCC subgroups and characterized subgroup survival differences, and (4) calculated correlations between image features and integrated biological pathways. To the best of our knowledge, this is the first study to extract HCC image features using pre-trained CNN models and assess correlations between image features and integrated pathways. Our results indicate the feasibility of applying CNN models to histopathological images to better understand disease diagnosis, prognosis, and pathophysiology.

## MATERIALS AND METHODS

### HCC datasets

We downloaded HCC histopathological images of diagnostic slides (access by TCGA-LIHC Diagnostic Slide Images) from the National Cancer Institute Genomic Data Commons Data Portal on 23 January 2019. In addition to images, this Portal also provides multiple molecular datasets (e.g., Transcriptomics, DNA Methylation, Copy Number Variation) and clinical information for the same cohort. In total, we obtained 966 H&E stained WSIs from 421 scanned HCC subjects (421 tumor samples and 105 normal tissue samples adjacent to tumors). The images were digitized and stored in .svs files, which contain pyramids of tiled images with differing levels of magnification and resolution. We used the Python modules OpenSlide and DeepZoomGenerator to read those image files. Most of the files contained three or four levels of sizes and resolutions, with level 4 corresponding to the highest resolution (median pixels: $89{,}640 \times 35{,}870$) and level 3 comprising 1/16 the size of level 4 (median pixels: $5{,}601 \times 2{,}249.5$). To reduce memory usage and processing time, we extracted either level 3 images or downsampled level 4 images (if available) by a factor of 16 to the level 3 equivalent. We removed two files which were either corrupted or did not contain level 3 or 4 information. In total, we used 964 files in our analysis.

We downloaded clinical files containing overall survival (OS) and disease free survival (DFS) information on 23 January 2019 from the cBioPortal for Cancer Genomics website (https://www.cbioportal.org/). The cBioPortal provides visualization, analysis and downloading of large-scale cancer genomics data sets. Importantly, cBioPortal includes data for the same patient cohort from which the HCC images were taken. When performing OS analysis, the event of interest is death (event = 1), while the censored event is being alive (event = 0). Thus, the number of days for event 1 and event 0 are the

number of days until death and number of days until last contact, respectively. In DFS analysis, the event of interest is new tumor occurrence (event = 1), while the censored event is the lack of detection of a new tumor (event = 0). In this case, the number of days for event 1 and event 0 are the number of days until detection of a new tumor and number of days until last contact, respectively.

We downloaded molecular pathway information, including integrated gene expression and copy number variation data, on 28 January 2019 from the Broad GDAC Firehose (https://gdac.broadinstitute.org/). This resource provides an open access web portal for exploring analysis-ready, standardized TCGA data including the cohort from which the TCGA-Liver HCC image files were collected. Using this pathway information, we applied the PAthway Representation and Analysis by Direct Inference on Graphical Models (PARADIGM) algorithm (*Vaske et al., 2010*) to infer Integrated Pathway Levels (IPLs). Briefly, PARADIGM predicts the activities of molecular concepts including genes, complexes, and processes and measures using a belief propagation strategy within the pathway context. Given the copy numbers and gene expression measurements of all genes, this belief propagation iteratively updates hidden states reflecting the activities of all of the genes in a pathway so as to maximize the likelihood of the observed data given the interactions within the pathway. In the end, the IPLs reflect both the data observed for that pathway as well as the neighborhood of activity surrounding the pathway. We used the analysis-ready file of IPLs generated by PARADIGM for correlation analyses between image features and biological pathways.

## Image pre-processing and feature extraction

For each of the 964 image files from 421 tumor to 105 normal samples, we performed stain-color normalization as described in previous image studies (*Macenko et al., 2009*; *Araújo et al., 2017*; *Rakhlin et al., 2018*). After color normalization, we performed 50 random color augmentations. We followed a previous study *Ruifrok & Johnston (2001)* and first deconvolved the original RGB color into H&E color density space. We then estimated a specific stain matrix for a given input image and multiplied the pixels with a random value from the range to obtain the color augmented image. We repeated the process to generate 50 augmentations. Next, we randomly selected 20 crops of size $256 \times 256$ and $512 \times 512$ pixels from each augmented image. We separately input each crop to the three pre-trained CNN models (VGG 16, Inception V3 and ResNet 50), each of which generated a total of 20 sets of features. Within each model, we combined all sets of features associated with an image into a single set by computing median values of features across all crops of all augmented images.

Deep CNN models such as VGG 16, Inception V3 and ResNet 50 contain millions of parameters that require extensive training on large datasets. When properly trained, these models have reached state-of-the-art performance in tasks such as image recognition and classification. To avoid the challenges of training an entire CNN from scratch, we used pre-trained versions of these models to extract histopathological image features in an unsupervised manner. This transfer learning approach was essential given the relatively small sample size of the HCC cohort. For the Inception and ResNet models, we used nodes

in the second-to-last convolutional layer as image features. For the VGG model, we concatenated nodes from the last four convolutional layers (block2_conv2, block3_conv3, block4_conv3 and block5_conv3) as image features. In each case, the CNN network weights had been pre-trained using ImageNet data (*Deng et al., 2009*). We implemented the above steps using Keras, a popular Python framework for deep learning.

## Sample visualization

To visualize samples, we first used Principal Component Analysis (PCA) to reduce the dimensionality of image features. We then applied the *t*-Distributed Stochastic Neighbor Embedding (*t*-SNE) method to visualize the first 10 components in 2 dimensions. The *t*-SNE method reduces data dimensionality based on conditional probabilities that preserve local similarity. We applied a *t*-SNE implementation that uses Barnes-Hut approximations, allowing it to be applied on large real-world datasets (*Van der Maaten & Hinton, 2008*). We set the perplexity to 15, and colored the sample points using the group information.

## Supervised classification from image features

We applied a linear Support Vector Machine (SVM) classifier (*Ben-Hur et al., 2001*) to discriminate between cancer and normal samples using the extracted image features (derived as described above). We used stratified 6-fold cross validation to train the model. To evaluate classifier performance, we visualized the Receiver Operating Characteristic (ROC) curve generated using cross-validation, with false positive rate on the $X$ axis and true positive rate on the $Y$ axis. We calculated the Area under the ROC curve (AUC) for each cross-validation fold, as well as the overall mean value. We also plotted the 2-class precision-recall curve to visualize the trade off between precision and recall for different prediction thresholds. A high AUC represents both high recall and high precision, which translates to low false positive and negative rates. Using average precision (AP), we summarized the mean precision achieved at each prediction threshold. We used the Python module Scikit-learn to perform classification with a linear SVM, setting the parameter C to its default value of 1.0.

## Survival analysis

To perform univariate survival analysis for each image feature individually, we applied Cox Proportional Hazards (CoxPH) regression models using the R package "survival" for both OS and DFS. We used a log-rank test to select significant image features with $p$-value $\leq 0.05$.

For multivariate survival analysis, we used the R package "glmnet" to build separate CoxPH OS models based on image features from each of the three pre-trained CNN models. We applied elastic net regularization with fixed alpha = 0.5, which corresponds to equal parts lasso and ridge regularization. In order to learn the optimal penalty coefficient lambda, we applied 10-fold cross validation. We evaluated models with the Concordance index (C-index) and a log-rank test. The C-index quantifies the quality of rankings and can be interpreted as the fraction of all pairs of individuals whose

predicted survival times are correctly ordered (*Pencina & D'Agostino, 2004*; *Steck et al., 2008*). A C-index of 0.5 indicates that predictions are no better than random.

### Subgroup discovery

Using the Scikit-learn Python module, we applied *K*-means clustering across all cancer samples to discover HCC subgroups. Specifically, we clustered all image features which were significantly associated with both OS and DFS. The *K*-means algorithm (*Lloyd, 1982*) clusters samples by minimizing within-cluster sum-of-squares distances for a given number of groups (K), which we varied between 2 and 12. To identify the optimal number of subgroups, we applied two metrics: the mean Silhouette coefficient and the Davies-Bouldin index. The Silhouette coefficient (*Rousseeuw, 1987*) takes values between −1 and 1, and it is calculated based on the mean intra-cluster distance and the mean nearest-cluster distance for each sample. Higher positive Silhouette values correspond to good cluster separation, values near 0 indicate overlapping clusters, and negative values indicate assignment of samples to the wrong cluster. The Davies–Bouldin index (*Davies & Bouldin, 1979*) is calculated based on the average similarity between each cluster and its most similar one, where an index close to 0 indicates a good partition. Given the optimal number of subgroups, we constructed CoxPH models to detect survival differences between the subgroups, again using C-index and log-rank test for evaluation. We fit Kaplan–Meier curves to visualize the survival probabilities for each subgroup.

### Correlation between image features and pathways

We calculated the Pearson correlation between image features and IPLs using the scipy Python module. Pearson correlation coefficients range between −1 and 1, with 0 implying no correlation. Each correlation coefficient is accompanied by a *p*-value, which indicates the significance of the coefficient in either the positive or negative direction. To correct for multiple hypothesis testing, we adjusted *p*-values using the Benjamini and Hochberg (BH) method (*Benjamini & Hochberg, 1995*). We selected significant correlations between image features and IPLs as those whose adjusted *p*-values were ≤ 0.05.

### Differential expression analysis

To identify differentially expressed (DE) pathways between two HCC subgroups, we applied the widely-used "Limma" R package (*Ritchie et al., 2015*). We selected significantly DE pathways as those whose Benjamini & Hochberg (BH)-adjusted *p*-values were ≤ 0.1.

## RESULTS

In this study, we made use of pre-trained CNN models VGG 16, Inception V3 and ResNet 50 to extract features from HCC histopathological WSIs. We first downsampled the WSIs, normalized the color, and generated augmented images. We then aggregated the features extracted from randomly selected crops using pre-trained CNN models. Using these image features, we performed survival analysis and subgroup discovery. We also performed correlation analysis between image features and integrated biological pathways. The workflow of these analysis steps can be seen in Fig. 1.

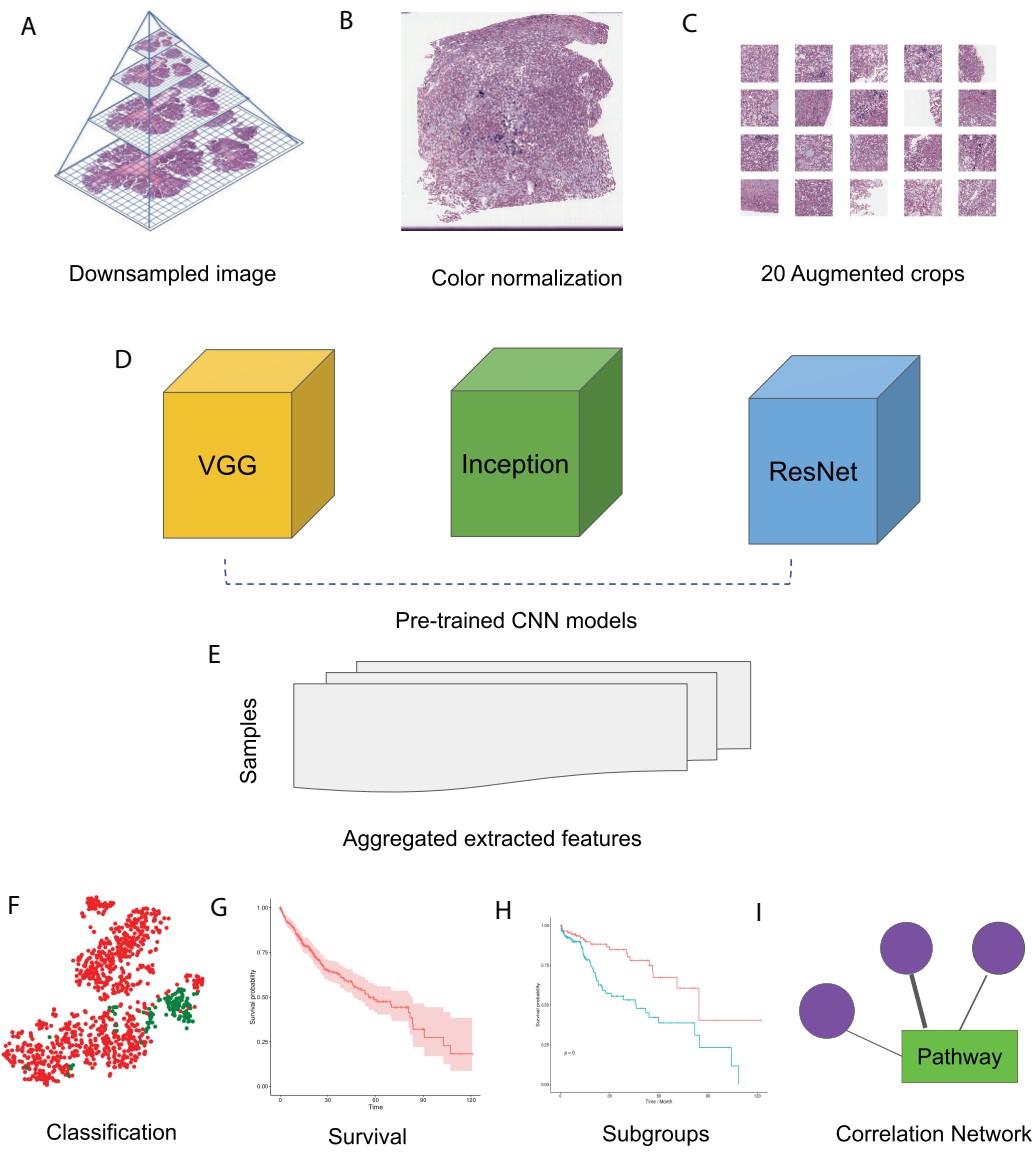

**Figure 1 HCC image analysis flow.** (A) For whole slide .svs files, downsampled images were generated, (B) color normalization was performed, (C) 50 augmented images were made for each original image and 20 crops were selected at random from each augmented image, (D) three CNN models, VGG 16, Inception 3 and ResNet 50 were applied to extract features from each crop, (E) features from all crops were aggregated and 50 sets of image features were obtained from each CNN model, (F) image features were used for classification, (G) image features were fit for survival analysis, (H) image features were used for subgroup discovery, (I) correlation between image features and biological pathways.

## Image feature extraction and survival analysis

Histopathology assessment is mandatory in HCC diagnosis (*Rastogi, 2018*), and the characteristics such as tumor number, size, cell differentiation and grade, and presence of satellite nodules were reported to be prognostic biomarkers (*Qin, 2002*). Given a histopathological image, CNNs enable efficient feature extraction and representation using convolutional, pooling, and fully connected network layers. To examine image

features relevant to HCC, we first downloaded HCC histopathological images from the National Cancer Institute Genomic Data Commons Data Portal. In addition to images, this Portal also provides multiple molecular datasets and clinical information for the same cohort of samples. We downloaded a total of 966 .svs image files from 421 cancer tissues and 105 tumor-adjacent normal tissues, of which 964 had sufficient information for the following analysis. For all image files, we used the equivalent of level 3 magnification (median 5,601 × 2,249.5 pixels) as described in the "Materials and Methods".

We performed staining color normalization, followed by image augmentation to improve sample variety. We randomly selected 20 crops of sizes 512 × 512 pixels or 256 × 256 pixels from each augmented image. The 20 512 × 512 crops represent 41.6% of the input image pixels on average, while the 20 256 × 256 crops represent 10.4% on average.

The deep CNN models VGG 16, Inception V3 and ResNet 50 contain millions of parameters, and extensive training of these models has led to state-of-the-art performance in image recognition and detection (*Rawat & Wang, 2017*). Given the small sample size in our cohort, we extracted features from each image crop by applying pre-trained versions of these models. This approach, which is a form of transfer learning, allows us to avoid the challenges of CNN model training from scratch. For the Inception and ResNet models, we chose all nodes in the second-to-last network layer as features after excluding the final fully-connected layers. For the VGG model, we chose all nodes from the last four convolutional layers as features. For each full image, we combined features from the 20 random crops into a single set of features representing that image.

In total, we obtained 1,408, 2,408 and 2,408 features for each image using the VGG 16, Inception V3 and ResNet 50 models, respectively. To aggregate these features across all augmented images, we computed median values for each feature. We then visualized cancer and normal samples in the context of these features by using PCA to reduce the feature dimensionality followed by applying *t*-SNE to the first 10 principal components. We also performed supervised classification of the samples using a linear SVM applied to each set of image features. Figure 2 shows these results using features derived from 256 × 256 crop sizes, with classification performance displayed as ROC and two-class precision-recall curves. The average AUC achieved by all three models is between 0.99 and 1, illustrating the clear separation achieved between tumor and normal samples using the extracted image features. Similarly, the AUCs achieved for features derived from 512 × 512 crop sizes were very close to 1. To compare this performance with that of an alternate method, we also applied PCA (randomized SVD) and SVD (full SVD) on the downsampled images without augmentation. Specifically, we extracted the first 100 principal components (PCA) or singular vectors (SVD) as features and performed supervised classification. Figure S1 shows that performance using PCA- and SVD-derived features is very poor. Finally, we performed classification on features derived without using image augmentation. Here, performance is only slightly worse, with AUCs ranging between 0.98 and 0.99 (Fig. S2).

We next compared the performance of a simpler network to that of the three CNNs evaluated in this study. Specifically, we applied a MobileNet v1 pre-trained network, which
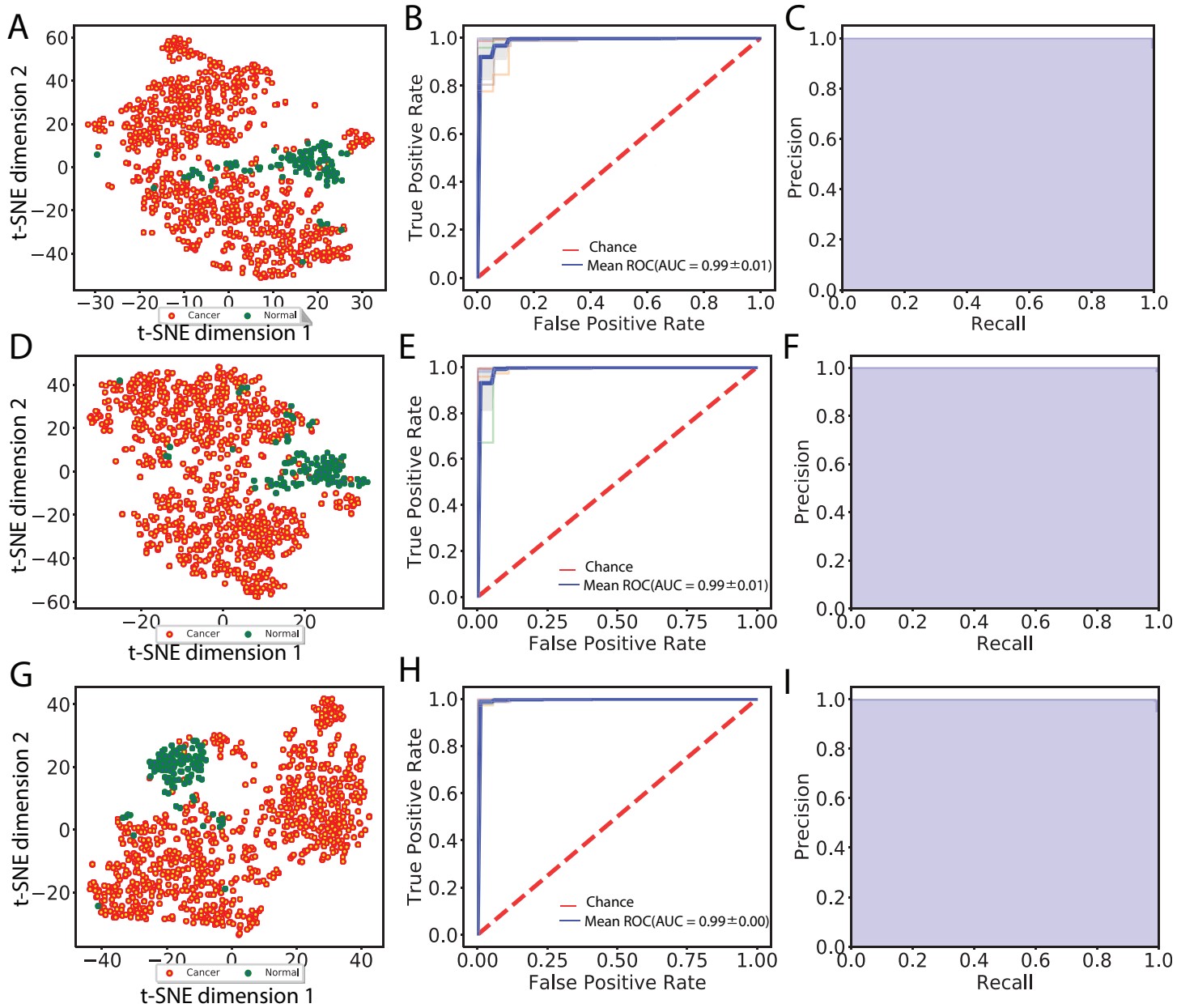

**Figure 2 Visualization of extracted image features and classification between cancer and normal samples.** (A), (D) and (G) indicate *t*-SNE visualization, (B), (E) and (H) indicate ROC curves from linear SVM and (C), (F) and (I) indicate recall and precision curves measured using VGG image features, Inception features and ResNet features, respectively.

has many fewer tunable parameters ($4.2 \times 10^6$) than VGG 16 ($1.4 \times 10^8$), Inception v3 ($2.4 \times 10^7$), and ResNet 50 ($2.3 \times 10^7$). As with the other networks, we removed the final layer of MobileNet v1 and used the network to extract features for each image. We aggregated these features as before, followed by performing SVM classification. We found that the classification performance using MobileNet v1 was indistinguishable from those achieved by the larger networks. This result suggests that the pre-trained

networks used in our study contain many more tunable parameters than are strictly necessary to yield very good classification performance.

We also explored reduction of model complexity by selecting smaller and smaller subsets of pre-trained CNN image features for classification. Figure S3 displays performance using randomly-selected image feature subsets of size 10, 25, 50 and 100 in each of the three pre-trained CNNs using 256 × 256 pixel crops. Our results show that when using smaller and smaller sets of features, classification AUC reached as low as 0.84, which was substantially worse than our original results. However, using random sets of 100 features led to performance that was nearly as good as that achieved using all features. Overall, our results show that use of CNN-derived image features is extremely effective for distinguishing HCC tumor from normal samples, which suggests that pre-trained CNN models capture the most relevant characteristics from HCC histopathological images.

To aid in interpreting CNN-derived image features of HCC, we visualized feature mappings of VGG model convolutional layer blocks when applied to 256 × 256 pixel crops of histopathological images (Fig. 3; Fig. S4). We note that the first convolutional layers tend to resemble the original image, but subsequent layers seem to intensify partial objects. In order to study whether the CNN-derived image features were associated with clinical survival, we next performed univariate CoxPH regression survival analysis on each feature. We obtained clinical information for each sample from the cBioPortal for Cancer Genomics, as described in the "Materials and Methods". For the subjects with multiple histopathological images, we computed median feature values across the images for the following survival analysis. For each image feature, we applied CoxPH regression models for both OS and DFS and selected significantly associated features ($p$-value ≤ 0.05) based on a Score (log-rank) test. We also validated the predictive ability of the survival models using C-index. Table 1 shows the number of significant features for each model and survival type. 21.4% and 16% of image features on average were significantly associated with OS and DFS, respectively. Each model had a slightly different number of significant features, with more features associated with OS than DFS.

Finally, we performed multivariate CoxPH regression analyses for each survival type on all image features from each model. We employed elastic net regularization using equally weighted lasso and ridge regularization during model training. Optimal hyperparameters were selected using 10-fold cross-validation and subsequently used for model prediction. Overall, we identified three multivariate OS models with the following log-rank $p$-values and C-indices: 1.2E−23 and 0.788 (VGG), 7.6E−18 and 0.789 (Inception), and 1.2E−12 and 0.739 (ResNet) from the 256 × 256 crop sizes. Table 2 displays the C-indices and $p$-values achieved for each pre-trained network, image crop size and survival type. The Inception-derived model achieved the highest indices of 0.789 at OS and 0.744 at DFS. Overall, our results show that CNN-derived image features are significantly associated with clinical survival and can be used to build accurate survival models.
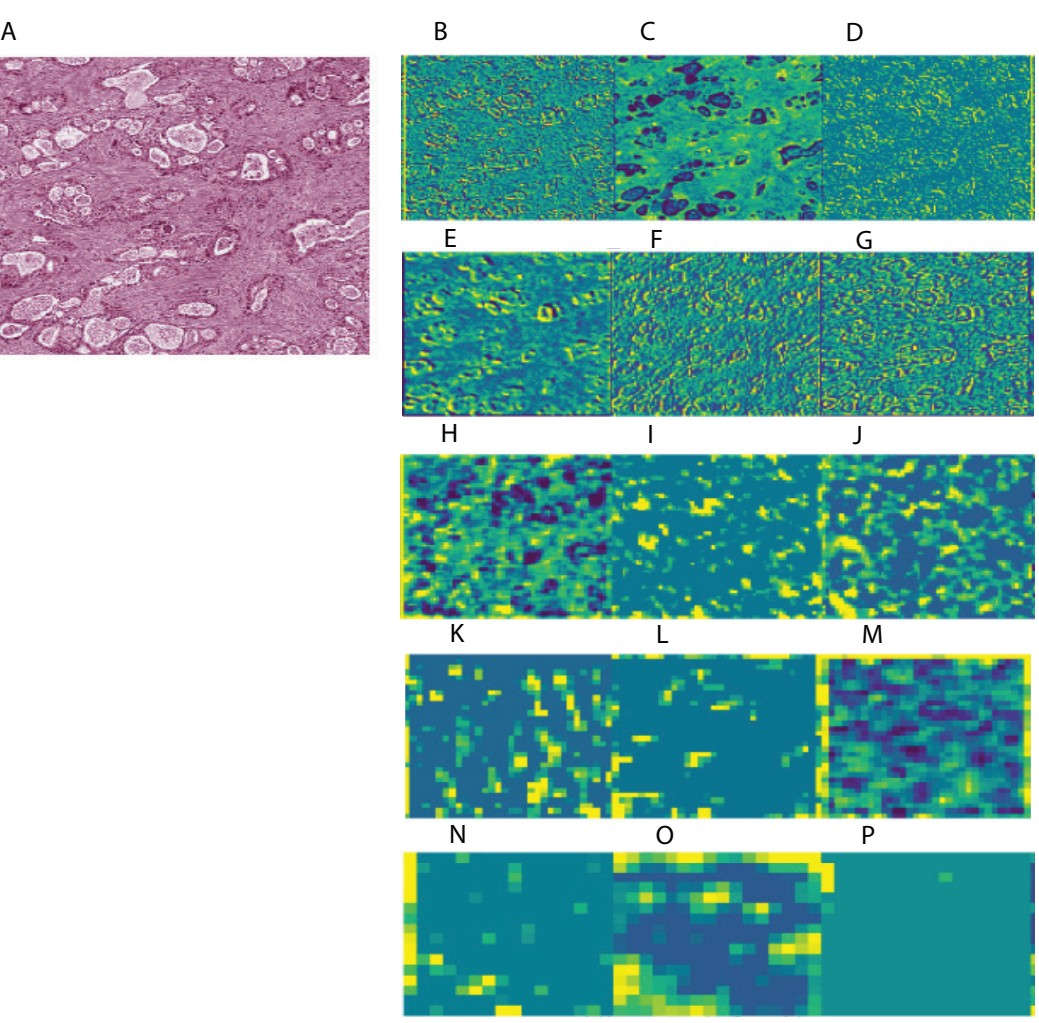

**Figure 3 Example of feature mapping visualization in VGG 16 model in one cancer sample.**
(A) shows an image patch with 256 × 256 pixels. (B)–(P) indicates the corresponding feature mapping from convolutional block 1 (B–D) to convolutional block 5 (N–P).

**Table 1 Significant image feature number from univariate CoxPH regression models.**

| Model | Feature number | Crop size | Significant features of OS | Significant features of DFS |
|---|---|---|---|---|
| VGG | 1408 | 256 | 272 (19.3%) | 219 (15.6%) |
| Inception | 2048 | 256 | 574 (28.0%) | 294 (14.4%) |
| ResNet | 2048 | 256 | 522 (25.5%) | 385 (18.8%) |
| VGG | 1408 | 512 | 300 (21.3%) | 201 (14.3%) |
| Inception | 2048 | 512 | 356 (17.4%) | 290 (14.2%) |
| ResNet | 2048 | 512 | 347 (17.0%) | 390 (19.0%) |

## Subgroup discovery from image features

To investigate whether our CNN-derived image features relate to HCC prognosis, we next used these features to discover subgroups within tumor samples. We considered all image

**Table 2 Multivariate CoxPH regression model in three models.**

| Model | Crop | Survival | C-index | P value |
|---|---|---|---|---|
| VGG | 256 | OS | 0.788 ± 0.022 | 1.2E−23 |
| Inception | 256 | OS | 0.789 ± 0.021 | 7.6E−18 |
| ResNet | 256 | OS | 0.739 ± 0.025 | 1.2E−12 |
| VGG | 256 | DFS | 0.655 ± 0.019 | 1.5E−08 |
| Inception | 256 | DFS | 0.744 ± 0.018 | 3.2E−13 |
| ResNet | 256 | DFS | 0.7 ± 0.019 | 4.1E−11 |

features which were significantly associated with both OS and DFS. Using these features, we clustered the tumor samples using $K$-means ($K = 2$–$12$) and used both Silhouette coefficients and Davies-Bouldin values to choose the optimal number of subgroups. As shown in Fig. 4, two subgroups were determined to be optimal for all three models. We visualized these subgroups using $t$-SNE to reduce dimensionality.

We then examined survival differences between the subgroups. For each model and survival type, we generated Kaplan–Meier survival curves stratified by subgroup. Our results (Fig. 5) note that the subgroups discovered using the Inception and ResNet models show a significant difference in both OS and DFS using a log-rank test. The two subgroups from Inception have the most significant OS difference, with $p$-value 7.39E−07 and C-index 0.628, followed by the two subgroups from ResNet with $p$-value 0.001 and C-index 0.582. We also observed significant differences in DFS between subgroups in both models, with $p$-values and C-indices of 0.012 and 0.558 (Inception) and 0.014 and 0.56 (ResNet), respectively. For the VGG model, we only detected a significant difference for DFS ($p$-value 0.007 and C-index 0.536). In all models, we note that the second subgroup ("group 2") has consistently better OS and DFS survival than the first subgroup ("group 1"). Table 3 shows the subgroup overlap between the three models. Overall, 176 samples from the Inception group 1 were also labeled group 1 in VGG and ResNet models. In contrast, 109 samples from the Inception group 2 were identified as group 2 in ResNet but group 1 in VGG. Taken together, the significant survival differences detected between sample subgroups demonstrate the feasibility of discovering clinically-relevant HCC subgroups using CNN-derived image features.

## Correlation between image features and biological pathways

Previous studies examined the molecular mechanisms underlying HCC (*Guichard et al., 2012*; *Totoki et al., 2011*; *Schulze et al., 2015*; *Ally et al., 2017*). To relate our CNN-derived image features to such mechanisms, we identified correlations between features and a collection of molecular pathways. Specifically, we first obtained IPLs using the Firehose Genome Browser, which provides analysis-ready files inferred from both gene expression and DNA copy number variation using the PARADIGM algorithm (*Vaske et al., 2010*). IPLs indicate the predicted activities of biological concepts using both copy number and gene expression data (described in "Materials and Methods"). The IPL matrix contains a total of 7,202 entities derived from 3,656 concepts in 135 merged pathways. Each entity is annotated with the concept (gene) and pathway index as shown by the example

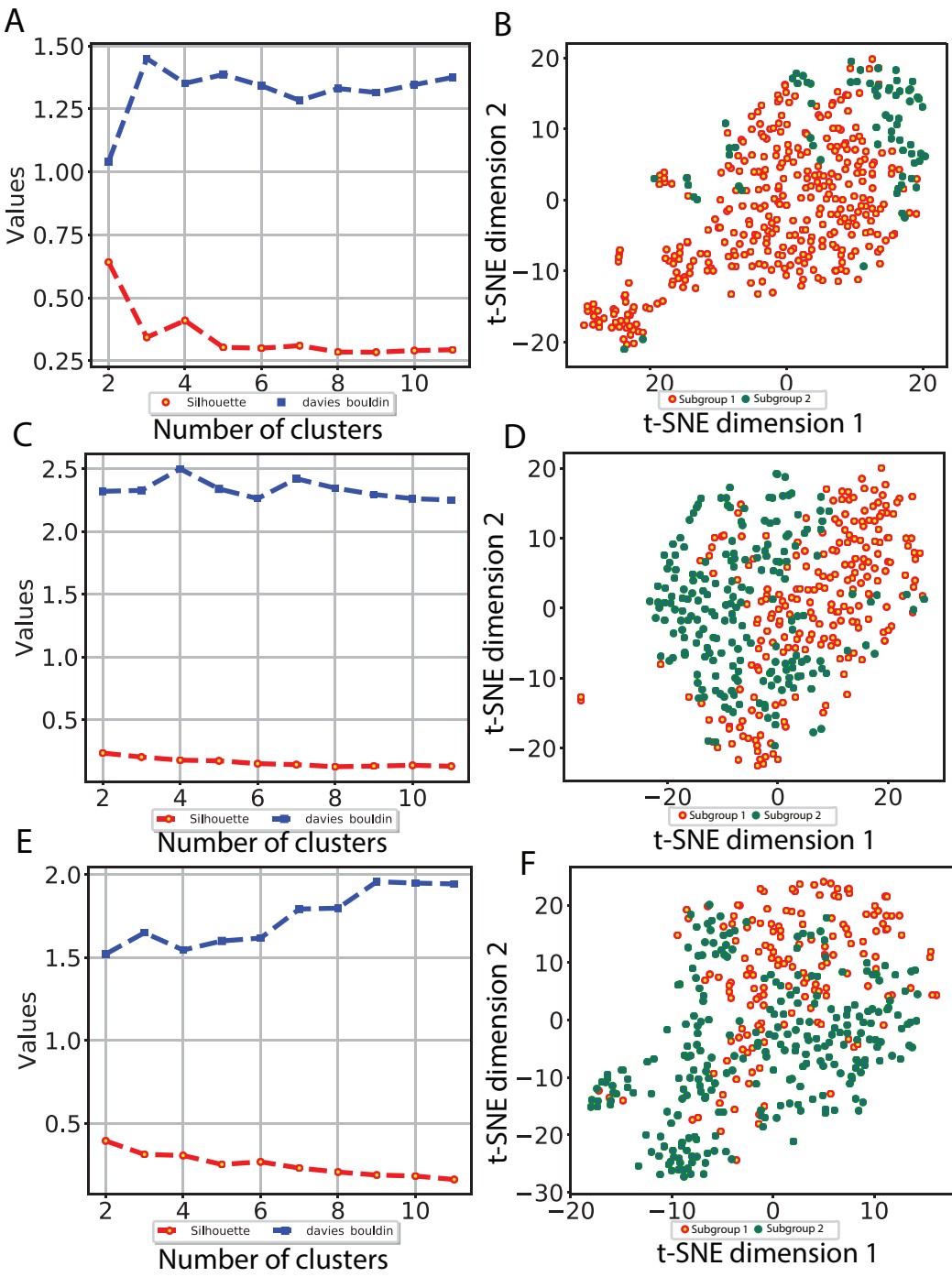

**Figure 4 Subgroup discovery from image features using 256 × 256 pixel crop size.** (A), (C) and (E) display two different metrics for selecting the optimal number of clusters, and (B), (D) and (F) indicate the *t*-SNE visualization of best clusters using VGG image features, Inception image features and ResNet image features, respectively.

19_EPHB3. Here, the EPHB3 gene participates in EPHB forward signaling whose pathway index is 19. We first computed Pearson correlation coefficients between these IPLs and each feature significantly associated with both OS and DFS. We then selected significantly

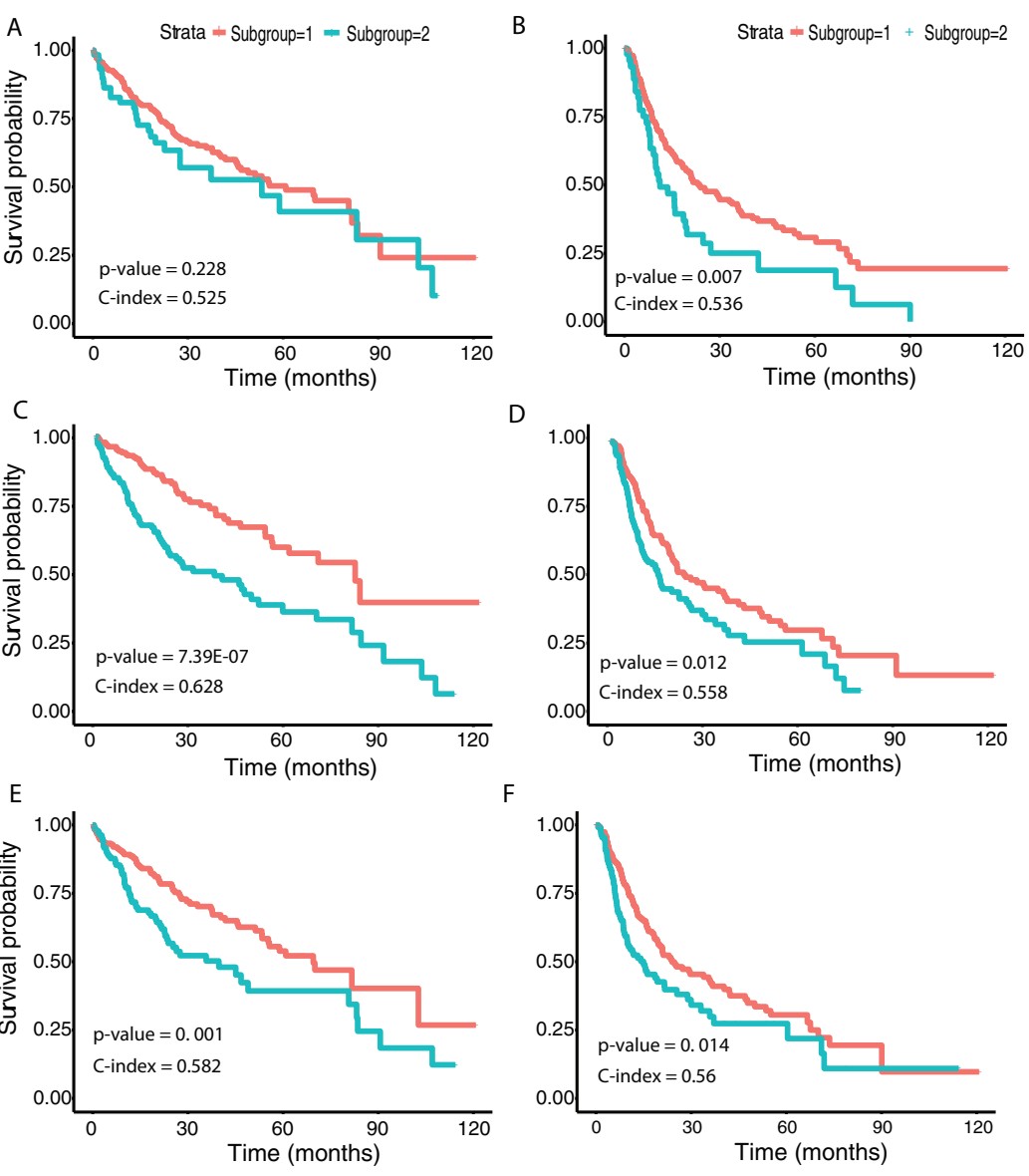

**Figure 5 Survival analysis from discovered subgroups.** (A), (C) and (E) correspond to the CoxPH model applied to OS, (B), (D) and (F) correspond to DFS. The two groups are indicated in red and green, using VGG image features, Inception image features and ResNet image features, respectively.

correlated IPL-feature pairs based on Benjamini and Hochberg (BH) (*Benjamini & Hochberg, 1995*)-adjusted *p*-values ≤ 0.05. With 256 × 256 crop sizes, 90 (out of 97), 199 (out of 203) and 192 (out of 203) survival-associated image features from the VGG, Inception and ResNet models, respectively, were significantly correlated with IPLs. On average, 90.2% of the image features showed a significant correlation, with Pearson correlation coefficients ranging between −0.536 and 0.385.

Finally, we performed differential expression analysis to identify IPL differences between each pair of sample subgroups. For each model, we selected pathways with

**Table 3 Overlaps of subgroup (1/2) frequency counts between three pre-trained CNNs.**

| Inception | VGG 16 | ResNet | Sample count |
|---|---|---|---|
| 1 | 1 | 1 | 176 |
| 1 | 1 | 2 | 18 |
| 1 | 2 | 1 | 20 |
| 1 | 2 | 2 | 4 |
| 2 | 1 | 1 | 48 |
| 2 | 1 | 2 | 109 |
| 2 | 2 | 1 | 16 |
| 2 | 2 | 2 | 30 |

BH-adjusted $p$-values 0.05. Surprisingly, we found no significant pathways at this threshold for all three models and both crop sizes. After relaxing the $p$-value threshold to 0.1, we detected five significant entities from two pathways: 19: EPHB forward signaling (EPHB3, ROCK1 and Ephrin B1/EPHB3) and 66: Glucocorticoid receptor regulatory network (IL8 and ICAM1). The two entities at pathway 66 were calculated between two subgroups from Inception model with 256 × 256 crops while the three entities at pathway 19 were from VGG model with 512 × 512 crops. Figure 6 shows a network visualization of these pathways with significantly-correlated image features. The nodes represent image features and pathways, while the thickness of the edges denote the observed Pearson correlation coefficients. The numbers on the image feature nodes were assigned according to the order from the initial feature extraction. We note that some image features showed correlation with more than one entity from the same pathway, while others seemed to be related to only one entity. Overall, 31 out of 49 image features with significant correlations were found using the Inception model, of which three features (324, 1,859 and 1,292) were correlated with pathway 19: EPHB forward signaling.

The VGG model identified a total of four significantly-associated features (two each of 870 and 871) from 256 × 256 to 512 × 512 crops. Feature 870 showed correlation with only 19: EPHB forward signaling, while feature 871 was correlated with both 19: EPHB forward signaling and 66: Glucocorticoid receptor regulatory network. The observation that consecutive features from the VGG model were correlated with similar pathways suggests that these features represent related attributes of the original images. In addition, it is noteworthy that the model with the largest proportion of significantly-associated features (Inception) also showed the most significant survival analysis results.

## DISCUSSION

In this study, we applied the pre-trained CNN models VGG 16, Inception V3 and ResNet 50 to extract features from HCC histopathological WSIs. Using these image features, we observed clear separation between cancer and normal samples both visually ($t$-SNE) and through supervised classification. By performing univariate CoxPH regression, we identified averages of 21.4% and 16.0% of image features significantly associated with OS and DFS, respectively. Many of these image features were also significantly associated with OS in a multivariate CoxPH regression model. We utilized the CNN-derived image

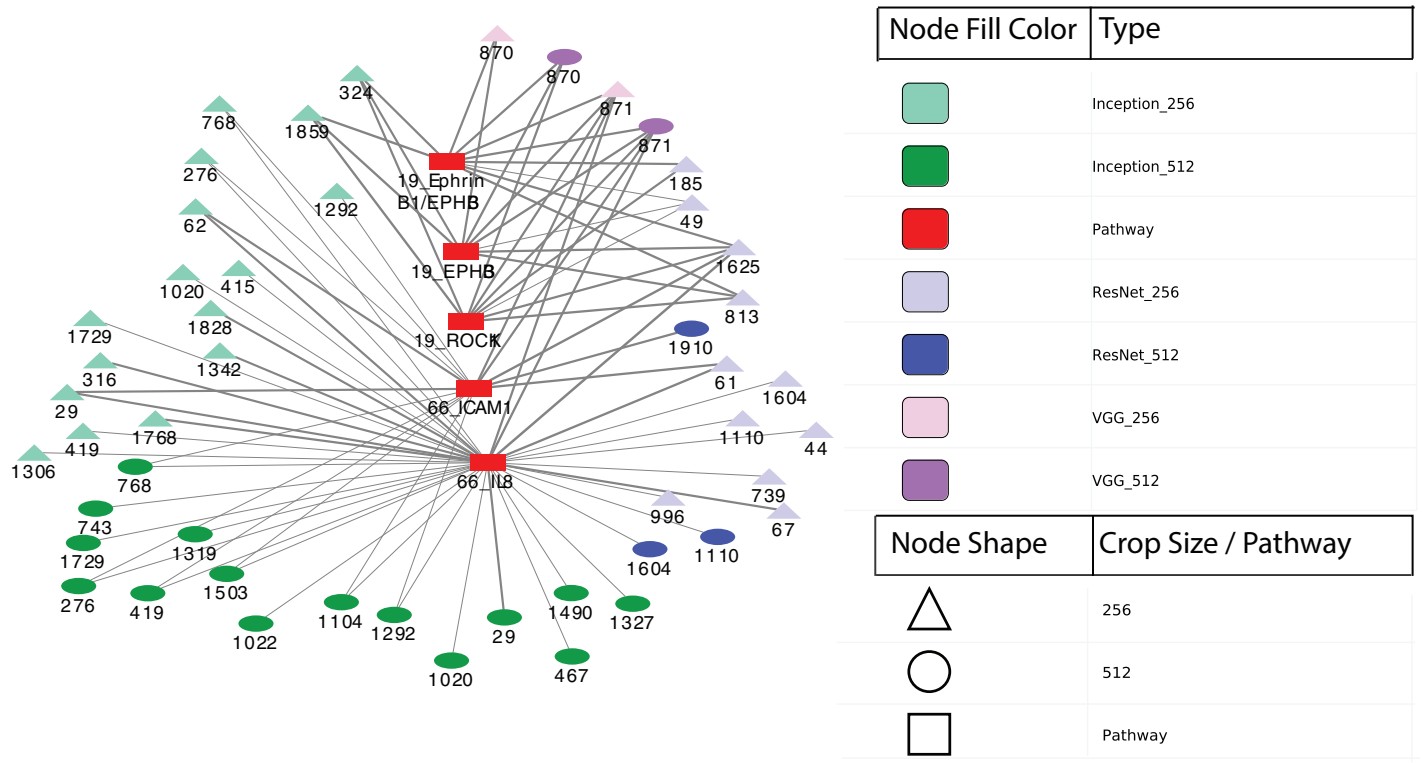

**Figure 6 Correlation network between image features and example pathways.** Colors of nodes indicate CNN models VGG, Inception and ResNet, as well as pathways. The labeled names of image features consist of the model name, crop size and feature order number. The thickness of each edge corresponds to the magnitude of correlation coefficients ranging between −0.536 and 0.385 that were statistically significant with the range.

features to discover HCC subgroups, with the optimal two subgroups showing a significant difference in both OS and DFS in the Inception model.

Notably, we found that 90.2% of the image features significantly associated with both OS and DFS were also significantly correlated with measures of IPLs. The five most significant IPL entities were found in two pathways—EPHB forward signaling and Glucocorticoid receptor (GR) regulatory network—implying a potential role for these pathways in the prognosis of HCC. Previous studies of EPHB forward signaling have shown that it induces cell repulsion and controls actin cell adhesion and migration (*Park & Lee, 2015*). It has also been reported that EPHB receptors and ephrin ligands are involved in carcinogenesis and cancer progression (*Xi et al., 2012*). Finally, the EPHB3 receptor also inhibits the Wnt signaling pathway (*Li, Chen & Chen, 2014*), which was reported to be useful for HCC stratification (*Ally et al., 2017*). In addition, previous studies have reported that the glucocorticoid receptor binds promoters, interacts with other transcription factors (*Le et al., 2005*), and causes HCC (*Mueller et al., 2011*) in mice when receptor signaling is impaired. GR regulatory network member Interleukin-8 (IL8), a proinflammatory CXC chemokine, was reported to promote malignant cancer progression (*Waugh & Wilson, 2008*), while member Intercellular cell adhesion molecule-1 (ICAM-1) has functions in immune and inflammatory responses and was reported to play a role
in liver metastasis (*Benedicto, Romayor & Arteta, 2017*). We note that a previous study performed integration of genomic data and cellular morphological features of histopathological images for clear cell renal cell carcinoma, finding that an integrated risk index from genomics and histopathological images correlated well with survival (*Cheng et al., 2017*). In addition, a second study (*Mobadersany et al., 2018*) developed a CNN model using both histopathological and genomic data from brain tumors, which surpassed the current state of the art in predicting OS.

Stratification of patients is an important step to better understand disease mechanisms and ultimately enable personalized medicine. Previous studies of HCC have suggested molecular-level subgroups (*Goossens, Sun & Hoshida, 2015*; *Hoshida et al., 2009*; *Chaudhary et al., 2017*). In the recent study, the authors applied deep learning to integrate three omic datasets from 360 HCC patients (the same cohort used in our study), discovering two subgroups with survival differences. In our work, we identified subgroups using all three CNN models, with the subgroups from both Inception (C-index = 0.628; $p$ value = 7.39E−07) and ResNet (C-index = 0.582; $p$ value = 0.001) models showing significant differences in OS. We note that this significance of the Inception model is lower than that achieved using subgroups identified using multiple omic data integration (C-index = 0.68 and $p$ value = 7.13E−6) (*Chaudhary et al., 2017*), although the C-index is also slightly lower. We also detected significant survival differences in DFS using all three models, which to our knowledge has not been previously investigated. Interestingly, the subgroups from Inception model were most significantly different in OS.

In the analysis of histopathological images, the large image size and different levels of resolution from WSIs pose challenges to accurate and efficient feature selection (*Komura & Ishikawa, 2018*). To avoid information loss, WSIs are often divided into small patches (e.g., 256 × 256 pixels) and each patch is analyzed individually as a Region of interest (ROI). These ROIs are first labeled using active learning (*Cruz-Roa et al., 2018*) or by professionally trained pathologists (*Nagarkar et al., 2016*). Subsequently, averaged regions of patches representing WSIs are studied for specific tasks (*Komura & Ishikawa, 2018*). In our work, we randomly selected 20 patches of 256 × 256 and 512 × 512 pixels from each WSI and extracted features from the last layers of CNN models to represent each image for visualization and classification. To robustly deal with color variation and image artifact issues, we conducted color normalization and augmentation before applying CNN models. Color normalization adjusts pixel-level image values (*Bejnordi et al., 2016*), and color augmentation generates more data by altering hue and contrast in the raw images (*Lafarge et al., 2017*). We achieved very good classification performance, with AUCs between 0.99 and 1 for distinguishing between normal and tumor samples. To illustrate the power of a transfer learning approach using pre-trained CNNs, we also applied a simple (not pre-trained) CNN model (Fig. S5) for classifying tumor and normal samples. This approach achieved a best validation accuracy of 87.8% (Fig. S6), which was substantially worse than the transfer learning performance.

Comparing our performance to previous work, we note that in one study of histopathological images (*Alhindi et al., 2018*), classification performance reached 81.14% accuracy using the extracted features from a pre-trained VGG 19 (similar to VGG 16)

network. In a similar study of histopathological images of breast cancer (*Rakhlin et al., 2018*), classification performance on 400 H&E-stained images of 2,048 × 1,536 pixels each reached an AUC of 0.963 for distinguishing between non-carcinomas vs carcinomas samples. We note that our study uses higher resolution histopathological images (median 5,601 × 2,249.5 pixels), which may explain the better performance.

Recent related work in histopathological image analysis include a deep-learning-based reverse image search tool for images called Similar Medical Images Like Yours (SMILY) (*Hegde et al., 2019*). By building an embedding database using a specialized CNN architecture called a deep ranking network, SMILY enables search for similar histopathological images based on a query image. SMILY's deep ranking network utilizes an embedding-computing module that compresses input image patches into a fixed-length vector. This module contains layers of convolutional, pooling, and concatenation operations. SMILY retrieves image search results with similar histological features, organ sites, and cancer grades, based on both large-scale quantitative analysis of annotated tissue regions and prospective studies with pathologists blinded to the source of the search results. SMILY's creators comprehensively assessed its ability to retrieve search results in two ways: using pathologist-provided annotations, and via prospective studies where pathologists evaluated the quality of SMILY search results.

Additional related work has made use of deep learning generative models to help delineate fundamental characteristics of histopathological images. Generative Adversarial Networks (GANs) have enjoyed wide success in image generation. GANs involve training a generator to fool a discriminator, while a discriminator is trained to distinguish the generated samples from real samples. This approach eventually produces high-quality images (*Goodfellow et al., 2014*). The creators of Pathology GAN recently demonstrated its abilities to create artificial histological images and learn biological representations of cancer tissues (*Quiros, Murray-Smith & Yuan, 2019*). A second type of generative model known as a variational autoencoder (VAE) learns the distribution of latent variables and reconstructs images. VAEs have been successfully applied in image generation (*Kingma & Welling, 2013*), and a specialized version known as a conditional VAE can be suitable for pathology detection in medical images (*Uzunova et al., 2019*).

We note that our study has several limitations, including limited interpretability of the most discriminative HCC image features and a lack of external validation datasets. We also did not address multiclass grading on the HCC samples, instead focusing on a binary classification. Despite using pre-trained CNN models for feature selection, our results may still be limited by the somewhat small and unbalanced sample sizes of our dataset. Additional studies on other independent data sets should be evaluated to further explore the correlation between deep learning-based extracted images, clinical survival and biological pathways. Future work will involve experimenting with other CNN models, as well as improving the biological interpretation of features from pre-trained models.

## CONCLUSIONS

The image features extracted from HCC histopathological images using pre-trained CNN models VGG 16, Inception V3 and ResNet 50 can accurately distinguish normal and

cancer samples. Furthermore, these image features are significantly correlated with clinical survival and biological pathways.

## ACKNOWLEDGEMENTS

The views and conclusions contained in this document are those of the authors and should not be interpreted as representing the official policies, either expressed or implied, of the Army Research Laboratory or the U.S. Government. The U.S. Government is authorized to reproduce and distribute reprints for Government purposes notwithstanding any copyright notation herein. The authors thank the High Performance Computing Center and the Computational Research on Materials Institute at The University of Memphis (CROMIUM) for providing generous computing resources for this research.

### Funding

Research was sponsored by the Army Research Laboratory and was accomplished under Grant Number W911NF-17-1-0069. The funders had no role in study design, data collection and analysis, decision to publish, or preparation of the manuscript.

### Grant Disclosures

The following grant information was disclosed by the authors:
Army Research Laboratory: W911NF-17-1-0069.

### Competing Interests

The authors declare that they have no competing interests.

### Author Contributions

- Liangqun Lu conceived and designed the experiments, performed the experiments, analyzed the data, prepared figures and/or tables, authored or reviewed drafts of the paper, and approved the final draft.
- Bernie J. Daigle Jr conceived and designed the experiments, analyzed the data, authored or reviewed drafts of the paper, and approved the final draft.

### Data Availability

HCC Image Project Code: https://bitbucket.org/daiglelab/hccimage/src/master/.

### Supplemental Information

Supplemental information for this article can be found online at http://dx.doi.org/10.7717/peerj.8668#supplemental-information.

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
