# Peer review of "Prognostic analysis of histopathological images using pre-trained convolutional neural networks: application to hepatocellular carcinoma"

_PeerJ, doi:10.7717/peerj.8668_

## Round 0.1 · original submission · Major Revisions

Your manuscript has been reviewed and requires modifications prior to making a decision. The comments of the reviewers are included at the bottom of this letter.

Reviewers indicated that the methods and discussion sections in the manuscript should be improved.

They also recommended extensive English editing. I agree with this evaluation and I would, therefore, request for the manuscript to be revised accordingly.

I would also like to suggest the following change: in the methods section, the access date of database and accession numbers of datasets should be given.

Reviewer 1 ·

Basic reporting

While I believe the underlying concepts of transfer learning was appropriately applied, I have several concerns about the study.

-Please clarify the building and training process of concatenated networks. In my experience, random forest has been generally used to integrate feature information. Could the authors compare the proposed aggregation method and random forest? "The possibility of the combination of OCT and fundus images for improving the diagnostic accuracy of deep learning for age-related macular degeneration: a preliminary experiment" described the random forest feature aggregation.

-Three subplots in Figure 3 look the same to each other. The differnece should be emphasized.

-The configuration of Table 2 looks abnormal. Please review this table to be clear.

Experimental design

-Recent deep learning studies have revealed CAMs or attention maps. If possible, demonstrate the feature map examples for HCC sampels. Is there new features for HCC? More specific demonstration for deep learning application to histopathology is needed becuase the figures and discussion about deep learnign are too general.

-Is there other clinical information such as age, gender, diabetes, and hypertension? The combination of clinical information and deep learning features will
improve the prediction accuracy in survival analysis.

-The comparison between the traditional training model (using ADAM optimizer) and pretrained model should be discussed.

Validity of the findings

-PARADIGM combined with deep learning should be explained in more detail. Is it the first research to use both image-based deep learning and pathway analysis in HCC samples? The combination of deep learning and PARADIGM was the most interesting part in this manuscript, however the novelty was not highlighted. Please strengthen the explaination about Figure 6, because it failed to demonstrate the meaning of this study. Which pathways were activated and were associated with deep learning features (in attention maps)?

-Please disccuss about recent AI researches in the histopathology fields, such as "Similar image search for histopathology: SMILY" from Google.

-Study limitations were not disccussed. No external validation, relatively small number of dataset, and no multiclass grading should be addressed.

Additional comments

Although this study contain a novel approach for HCC samples, the manuscript looks old-fashioned. I recommend a more comprehensive exploration about interpretability in histopathological images and pathyway analysis.

·

Basic reporting

1. Although this paper is generally well written, further proofing is required. For instance, text in lines 52-53 and in 165 is nonsensical.

2. Figure 1 could be much clearer and better presented. Moreover, for other figures some axis labels are missing. The text to the far right in Figure 6 appears to be misaligned.

3. Table 1 and 2 require re-formatting. Table 2 appears to be incomplete.

4. I would suggest segregating the Background section into two sections entitled Introduction/Background and Related Works.

Experimental design

1. VGG-16, Inception V3 and ResNet-50 are quite large CNNs, and as I’m sure you are aware, they are quite computationally expensive. An ablation study is necessary to determine if simpler networks can be used to achieve similar performance. It’s difficult to deduce if the models used are well-fitting. Does an increase in regularization (by means of drop-out or by other regularization techniques) decrease or increase performance?

2. It is not entirely clear how the (89640 x 35870) and (5601 x 22495) images were down-sampled and randomly cropped to (256 x 256) and (512 x 512). During the down-sampling process, was the aspect ratio of the input images preserved? Inputs to VGG-16, Inception V3 and ResNet-50 commonly have a size of (3 x 224 x 224). At what image-preprocessing stages were images down-sampled and/or cropped, and what image sizes were fed into each network?

3. The image augmentation steps need to be more clearly detailed. How exactly was strain-color optimization performed? What exact modifications were made to images to augment their color? To what extent does image augmentation affect performance? I would suggest comparing performance with and without further image-augmentation.

4. The authors state that ‘We used these models to extract features in an unsupervised manner to avoid the challenges of training an entire CNN model scratch’. I believe that transfer learning could easily be used to improve the performance of the networks used- if this is to be left to future works, I suggest that this is stated transparently. While the usage of transfer learning is briefly discussed in the Discussion section, the authors do not make mention of Generative Adversarial Networks (GANs) or other systems that can be used to generate synthetic images.

5. How was the C parameter of the linear Support Vector Machine (SVM) determined?

6. ResNeXt has been demonstrated to obtain better results on both the ImageNet-5K set and the COCO detection set than its ResNet counterpart. Is there any reason as to why ResNet was used instead of ResNeXt?

Validity of the findings

No comment. Based solely on the approaches and methodology used, the findings are well stated, robust, statistically sound, controlled, and adequately linked to the research aims.

Additional comments

This paper uses pretrained Convolutional Neural Networks (CNNs) to extract features from Hepatocellular Carcinoma (HCC) histopathological images, and makes attempts to correlate them with relevant biological pathways. I would like to commend the authors on their efforts. The paper is interesting and well-written; however, I believe my questions and concerns should be addressed prior to publication.

---

## Round 0.2 · Minor Revisions

The manuscript has been re-assessed by the two reviewers, and they agree on the fact that there are still a few points that need to be addressed. We would be glad to consider a revision of your work, where the reviewers' comments will be carefully addressed one by one.

Reviewer 1 ·

Basic reporting

I appreciate the authors for the effort to revise manuscript.

Experimental design

no addtional comments

Validity of the findings

fig4 & 5
groups were described as "0" and "1".
they should be corrected.

Additional comments

The title is too broad.
I think "HCC" should be commented.

·

Basic reporting

The authors' have addressed all my comments.

Experimental design

I would like to acknowledge the authors' efforts towards addressing my comments, however, I still have some concerns:

1. An ablation study is still required- do simpler and/or smaller pretrained CNNs perform similarly to VGG-16, Inception V3 and ResNet-50?
2. Pretrained ResNeXt models for Keras are easily accessible: https://github.com/qubvel/classification_models. I suggest that the authors use either ResNeXt-50 or ResNeXt-101 instead of ResNet-50, which should improve performance.
3. Why was ResNet-50 used instead of ResNet-18, 24, 101 or 152? Does it exhibit better performance?

Validity of the findings

The findings are well stated, robust, statistically sound, controlled, and adequately linked to the research aims.

Additional comments

I would like to acknowledge the author’s efforts towards addressing my comments. Although the manuscript has significantly improved, I still have some concerns.

---

## Round 0.3 · accepted · Accept

The authors addressed the reviewers' concerns and substantially improved the content of MS. So, based on my own assessment as an academic editor, no further revisions are required and the MS can be accepted in its current form.

Reviewer 1 ·

Basic reporting

I appreciate the authors for revision and codes. Based on this research, many researchers will be inspired for histopathological image analysis.

Experimental design

MobileNet is a good option for the light-weight model.
No further comment.

Validity of the findings

No further comment.

Additional comments

No further comment.